# Combined Cytokine Blockade Therapy (CCBT) Using Basiliximab and Infliximab for Treatment of Steroid-Refractory Graft-Versus-Host Disease (SR-GvHD)

**DOI:** 10.3390/cancers16233912

**Published:** 2024-11-22

**Authors:** Hoda Pourhassan, Tina Nguyen, Dongyun Yang, Salman Otoukesh, Shukaib Arslan, Amanda Blackmon, Vaibhav Agrawal, Idoroenyi Amanam, Brian Ball, Paul Koller, Amandeep Salhotra, Ahmed Aribi, Pamela Becker, Peter Curtin, Andrew Artz, Ibrahim Aldoss, Haris Ali, Forrest Stewart, Eileen Smith, Anthony Stein, Guido Marcucci, Stephen J. Forman, Ryotaro Nakamura, Monzr M. Al Malki

**Affiliations:** 1Department of Hematology and Hematopoietic Cell Transplantation, City of Hope National Medical Center, Duarte, CA 91010, USA; sotoukesh@coh.org (S.O.); sarslan@coh.org (S.A.); ablackmon@voh.org (A.B.); vagrawal@coh.org (V.A.); iamanam@coh.org (I.A.); brball@coh.org (B.B.); pkoller@coh.org (P.K.); asalhotra@coh.org (A.S.); aaribi@coh.org (A.A.); pbecker@coh.org (P.B.); pcurtin@coh.org (P.C.); aartz@coh.org (A.A.); ialdoss@coh.org (I.A.); harisali@coh.org (H.A.); fstewart@qcoh.org (F.S.); esmith@coh.org (E.S.); astein@coh.org (A.S.); gmarcucci@coh.org (G.M.); sforman@coh.org (S.J.F.); rnakamura@coh.org (R.N.); malmalki@coh.org (M.M.A.M.); 2Pharmacy, City of Hope National Medical Center, Duarte, CA 91010, USA; tinanguyen@coh.org; 3Department of Computational and Quantitative Medicine, Division of Biostatistics, City of Hope National Medical Center, Duarte, CA 91010, USA; donyang@coh.org

**Keywords:** basiliximab, infliximab, steroid refractory, graft-versus-host disease

## Abstract

The standard first-line treatment for acute graft-versus-host disease (aGvHD) is systemic, high-dose glucocorticoids which often have inadequate response rates requiring next-in-line therapies. Ruxolitinib is commonly used in these cases; however, patients can be ruxolitinib-intolerant or -refractory, necessitating further interventions. Here, we discuss the use of combined cytokine blockade therapy (CCBT) using the monoclonal antibodies infliximab (a TNF-α inhibitor) and basiliximab (an IL-2 receptor blocker), which, though often used clinically, have not been adequately explored in the literature. This study evaluated the overall response rate, non-relapse mortality, and overall survival of CCBT in a cohort of steroid-refractory aGvHD patients and finds that CCBT could serve as an acceptable alternative when patients are ruxolitinib-intolerant.

## 1. Introduction

Allogeneic hematopoietic cell transplantation (HCT) is a curative option for patients facing life-threatening hematologic disease. Graft-versus-host disease (GvHD) is a leading cause of non-relapse mortality (NRM) following HCT in both HLA-matched sibling and unrelated donor settings [1,2]. Advancements in GvHD prophylactic regimens and post-HCT supportive care measures have resulted in improved survival following HCT. However, clinically significant acute GvHD (aGvHD) still develops in approximately 30–50% of HCT recipients [3,4] and often results in increased transplant-related morbidity and mortality and deterioration of quality of life after HCT [5].

The standard first-line treatment for aGvHD is systemic, high-dose glucocorticoids [5,6], but unfortunately, the response rate to first-line therapy is not robust with grade II disease observed in approximately 60% of patients and grade IV disease in 30 to 40% of patients [7,8]. Furthermore, glucocorticoids are associated with clinically significant side effects including susceptibility to infection and other long-term complications, which result in a lower quality of life and an adverse effect on survival [9]. Until the recent phase II and III REACH trials evaluating the role for ruxolitinib (an oral JAK inhibitor), there has been little consensus regarding streamlined second-line treatment and subsequent lines of treatment for patients with steroid-refractory aGvHD (SR-GvHD), and outcomes remain poor [10,11]. Several immunosuppressive drugs have been proposed for patients with SR-GvHD, the studies for which are small, retrospective, uncontrolled, and use different endpoints, making comparison and standardization difficult.

Infliximab (INF), a murine-human chimeric monoclonal antibody (mAb) that binds to both the soluble subunit and the membrane-bound precursor of tumor necrosis factor-alpha (TNF-α), has demonstrated clinical activity in the treatment of SR-GvHD by way of both small and large case series [12,13,14,15]. Basiliximab (BAS) is a murine-human chimeric monoclonal antibody which blocks the alpha-chain of the interleukin-2 (IL-2) receptor complex expressed on activated T lymphocytes known to be a critical pathway for activating cell-mediated allograft rejection. Though its single-agent use is not as prominent as infliximab in the literature, it too has shown activity in aGvHD in small case series [16,17,18]. The utilization of both agents in combination for SR-GvHD is often implemented clinically but has had limited discussion in the literature.

Combined cytokine blockade therapy (CCBT) using BAS/INF is an appealing intervention as it is parenteral and easy to administer compared to oral therapies, which is particularly important in gastrointestinal (GI) GvHD, where drug absorption may be an issue. It also does not incur clinically limiting cytopenia observed with other interventions such as ruxolitinib and extracorporeal photopheresis. Avoiding cytopenia is substantially important in the first 30–50 days post-transplant when graft dysfunction is common. Given the limited review of this combination in the available literature and its potential utilization in a clinical setting, we completed a retrospective analysis of outcome and response in patients with SR-GvHD treated with CCBT using BAS/INF.

## 2. Methods

### 2.1. Patient Population

The patients included were of any age in the range of 5–78 years, and had undergone HCT using any donor or graft source with any conditioning regimen between 2010 and 2021 at City of Hope Medical Center in Duarte, CA. The patients had steroid-refractory or steroid-dependent aGvHD, according to Mount Sinai Acute GvHD International Consortium guidelines [19] and received treatment with at least one dose of BAS and at least one dose of INF such that every patient received combination cytokine blockade therapy. Patients with >1 allogeneic transplant, chronic GvHD (cGVHD) including overlap syndrome, relapse of primary disease, graft loss, and glucocorticoid treatment for indications other than GvHD were all excluded (Figure 1). The study was conducted in accordance with the Helsinki Declaration and was approved by the institutional Investigational Review Board (City of Hope IRB 22177).

The primary objective was to ascertain a descriptive retrospective analysis for the use of CCBT using BAS/INF in SR-GvHD by way of overall response rate (ORR) at days 7, 14, and 28 for CCBT. The secondary objectives were to assess CR or PR to therapy calculated from the start of CCBT, which was defined as the date of administration of one dose of both BAS and INF, NRM at 6 months, and overall survival (defined as the time from CCBT to death from any cause) at 12 months. Patients who died, had progression of the underlying malignancy, or stopped treatment before day 28 were considered non-evaluable. Descriptive analyses were performed to summarize clinical and demographic characteristics. OS was estimated using the Kaplan–Meier method. All outcomes were assessed from the time of starting CCBT.

### 2.2. Treatment

BAS was administered intravenously as 20 mg doses given first as loading doses on days 1 and 4. Subsequent doses were given weekly starting 7 days after the final loading dose for 4–6 additional weeks but was continued per physician discretion. INF was administered as 10 mg/kg doses infused weekly until aGVHD progression or at the physician’s discretion. Tacrolimus, sirolimus, and cyclosporine dosing and monitoring were performed in accordance with institutional policies. Trough levels were monitored twice weekly to keep target tacrolimus and sirolimus levels between 5 and 10 ng/mL and cyclosporine levels 100–300 ng/mL. All patients received medications for veno-occlusive disease prophylaxis and anti-infective prophylaxis—including viral, fungal, and Pneumocystis jiroveci (PJP)—per institutional protocols.

### 2.3. Definitions

All patients had clinical presentation of SR-GvHD. SR-GvHD was defined as ≥1 of the following: GvHD increasing in stage in any organ or developing in a new organ after 3 days of ≥2 mg/kg methylprednisolone (MSPE) or equivalent, GvHD that has not improved in stage in ≥1 organ after 7 days of ≥2 mg/kg MSPE or equivalent, development of GvHD in a new organ after ≥1 mg/kg MSPE or equivalent for skin GvHD, or patients who progress during tapering before a 50% decrease in glucocorticoids is achieved.

Acute GvHD was staged and graded as per the consensus criteria [20], and the standard definitions were used to assess response [8,21]. CR was defined as the absence of any symptoms related to aGvHD; PR was defined as the improvement of at least one stage in the severity of aGvHD in one organ without deterioration in any other organ; and treatment failure was defined by the absence of improvement of aGvHD, deterioration of aGvHD in any organ by at least one stage, the development of aGvHD manifestations in a previously unaffected organ, or the use of any additional agents to control the disease.

Mortality cause was described as infection vs. GVHD depending on the steroid dose at the time of death. If the steroid dose > 20 mg prednisone or equivalent, the cause was attributed to GVHD; if the dose <20 mg prednisone or equivalent, the cause was attributed to infection.

### 2.4. Statistical Analysis

Descriptive statistics were used for baseline characteristics. Responses to therapy, as binary endpoints, were summarized as percentages. Kaplan–Meier curves and cumulative incidence curves were used for time-to-event endpoints and competing risk event endpoints, respectively. The associations between characteristics and responses to therapy were examined using a contingency table and chi-square tests. Log-rank and Gray’s tests were used for differences in time-to-event and competing risk event endpoints, respectively.

All tests were two-sided at a significance level of 0.05.

## 3. Results

Sixty patients who met the inclusion criteria were analyzed. The median age was 53 years (range: 5–78) at the time of HCT. Of 60 patients, 65% were male, and 18.3% of these male patients received grafts from female donors. The majority of patients were white (27/60, 45%) or Hispanic (20/60, 33%). The most common underlying hematologic diagnosis was acute myeloid leukemia (21/60, 35%) followed by myelodysplastic syndrome or myeloproliferative neoplasm (14/6, 23%) and acute lymphoblastic leukemia (9/60, 15%). The Karnofsky performance status at the time of HCT was 80–100% in 95% of patients. The HCT-CI score at the time of HCT was 3 or more in 47% of the patients. The conditioning regimen was ablative in 43% of patients with 77% of patients receiving PBSC grafts from mostly MSD/MUD in 75% of patients. It should be noted that among the GVHD regimens outlined, post-transplant cyclophosphamide is a more recent regimen used in practice, as the data collected spanned over a 10-year course. The baseline characteristics at time of transplant are described in Table 1.

The median time to the start of basiliximab following steroid therapy was 7 days with a range of −13–63 days, as one patient initiated basiliximab prior to treatment dose of MPSE. The time to the start of infliximab following steroid therapy was 11 days with a range from 0 to 150 days. Ruxolitinib was initiated in 33% (20/60) of patients prior to CCBT. Most patients had grade 3 or 4 overall aGvHD (grade 3: n = 35, 58%; grade 4: n = 20, 33%) at the time of CCBT initiation. GvHD characteristics are described in Table 2.

### 3.1. CCBT Response

The ORR for CCBT at days 7, 14, and 28 were 28.3% (17/60; CR 5.0%/PR 23%), 38.3% (23/60; CR 11.3%/PR 27%), and 38.3% (23/60; CR 23.3%/PR 15%), respectively. Ruxolitinib was initiated in 33.3% (20/60) of patients prior to CCBT. Patients who received ruxolitinib prior to CCBT had lower ORR at 25% (CR = 15%/PR = 10%) compared to those who did not at 47.5% (CR = 27.5%/PR = 20%) (Figure 2). Multivariate analysis was also completed and the only statistically significant characteristic impactful of ORR was female-to-male transplant with an ORR of 63.6% (CR 36.4%, PR 27.3%) vs. non-female-to-male transplants with an ORR of 32.7% (CR 20.4%, PR 12.2%) (Appendix A).

### 3.2. Survival Outcomes

At a median follow-up of 39.6 months (range: 12.1–71.9) among survivors, thirty-seven patients died—thirty-two of aGvHD, four of infection, one of sudden cardiac death, and one of leukemia relapse. The cumulative incidence of NRM was 51.7% (95% confidence interval, 38–64) at 6 months (Figure 3A). NRM was 47.5% (95% confidence interval, 31–62) at 6 months for those without ruxolitinib prior to CCBT and 60% (95% confidence interval, 34.5–78) for those with ruxolitinib prior to CCBT (Figure 4A). OS was 36.5% at 12 months (95% confidence interval, 26–49) (Figure 3B). OS was 40% (95% confidence interval, 25–55) at 12 months for those without ruxolitinib prior to CCBT and 30% (95% confidence interval, 12–50) for those with ruxolitinib prior to CCBT (Figure 4B). The cumulative incidence of any cGvHD was 33.6% at 12 months while it was 30% for extensive cGvHD at the same 12-month interval.

Multivariate analysis was also completed and a statistically notable impact on NRM was observed with myeloablative conditioning having lower NRM at 42% vs. 61% (*p* value 0.008) for non-myeloablative and reduced intensity conditioning regimens as well as lower NRM in matched donors at 47% vs. 71% (*p* value 0.013) in all other donor sources. For OS, the disease risk index for non-malignant diseases showed an OS of 45% in low or intermediate categories compared to15% (*p* value 0.042) in high and very-high-risk categories. Myeloablative conditioning regimens had an OS of 50% vs. 27% in non-myeloablative and reduced intensity regimens (Appendix A).

## 4. Discussion

This is a retrospective study and descriptive analysis of sixty HCT patients who received CCBT with BAS/INF in SR-GvHD, the results of which suggest potential efficacy of this combination regardless of prior ruxolitinib exposure. Our patient population spanned the spectrum of hematologic malignancies and overall had good performance status. However, the HCT-CI score at the time of HCT was 3 or more in almost half of the patients, which predicts both higher NRM and decreased survival [22,23]. Almost half of patients received ablative conditioning regimens and over three-fourths received PBSC grafts from predominantly MSD/MUD, each of which pre-emptively confer a higher risk of aGVHD [24,25].

More importantly, most patients had high-grade (3 or 4) or severe aGvHD at the time of CCBT initiation, with aGVHD predominantly involving the skin and GI organs at presentation. GI GvHD was overall more severe (grade 3 and 4), reflecting our center’s practice in using this combination with GI GVHD [26]. The incidence of severe GI GVHD (stage 3–4) has decreased during the past decade; however, treatment remains unsuccessful in most cases, and the GI tract is involved in virtually all fatal cases of aGVHD [27,28]. Until the recent phase II and III REACH trials [10,11], which evaluated the role of ruxolitinib in the second-line clinical setting, there was little consensus regarding streamlined treatment, and outcomes remained poor. Notably, CCBT was our institution’s standard second-line therapy before the REACH trial data.

The response rates of infliximab and basiliximab as monotherapy have been previously studied. In one of the larger series involving a 52-patient panel, 71% of whom had grade III-IV aGVHD, only 15% achieved complete remission (CR) with the use of infliximab alone as salvage therapy [29]. In a 38-patient study of basiliximab alone, 21% of patients achieved a complete response while 58% had no response or disease progression [18].

In our CCBT data, CR rates are both comparable and more robust; although ORR on days 14 and 28 were the same, CR rates were higher on day 28, indicating a deeper response obtained with time and the low likelihood of response in patients who did not achieve at least PR by day 14. Patients who received ruxolitinib prior to CCBT had inferior ORR (25%) compared to those who did not (47.5%). In the REACH trials of ruxolitinib, the primary end point of overall response at day 28 observed for 39 patients was 55% [10], and though we cannot make direct cross-trial inference, these results are comparable to our ORR in patients who did not receive ruxolitinib prior to CCBT. While ruxolitinib has been adopted as the standard second-line therapy, its use is associated with resistance or intolerance in 1/5 of patients [30], and those with higher-grade aGVHD, particularly involving the GI tract or liver, appear to be less likely to respond [7,11]. In the REACH II trial, 32% of patients randomized to ruxolitinib therapy for SR-aGVHD discontinued treatment due to ineffectiveness or toxicity [4,29], and in a multi-center review of 48 ruxolitinib-resistant aGVHD patients, the overall response rate to subsequent therapy was 36%, further highlighting the possible role of CCBT in this setting [30]. To this point, amongst our study cohort, those who received and were refractory to ruxolitinib, 25% of patients were still able to be salvaged with CCBT.

We noted a statistically notable impact on NRM with myeloablative conditioning and matched donors having lower NRM than non-myeloablative or reduced intensity regimens and all other donor sources, which could perhaps indicate overall younger and more fit patients having a higher resiliency to tolerating prolonged illness with GvHD. In general, in SR-GvHD, NRM is high, and OS has historically been quite poor, with reported NRM of 63% at 18 months and average 6-month and 2-year OS rates of 49% and 17%, respectively [4,7]. Furthermore, studies of other T-cell-targeting agents in second-line therapy have not been able to mitigate this high NRM rate [21,31]. The use of second-line ruxolitinib in REACH I resulted in 1-year NRM and OS rates of 53% and 62%, respectively, and notably, patients who responded at day 28 and maintained their response had a far superior NRM rate compared to those who did not respond (28% vs. 84% at 1 year) [3,11]. In our CCBT study group, the overall NRM and OS at 1 year were 55% and 36.5%, but notably, both NRM (65% vs. 50%) and OS (30% vs. 40%) were inferior in those who received ruxolitinib prior to CCBT in comparison to those who did not. This again supports the notion that CCBT can act as salvage therapy in ruxolitinib-refractory patients given poorer outcomes in those who do not respond by day 28 and further supports for BAS/INF CCBT to be an acceptable second-line alternative therapy.

While we focused on the combination of BAS/INF for CCBT, other similar T-cell-targeting antibodies have been implemented in the literature. In the phase 2 study of corticosteroids combined with natalizumab, a humanized mAb that blocks T-cell trafficking to the GI tract, the antibody failed to have impact on clinical response or to improve the outcomes of NRM or OS in patients with newly diagnosed high-risk GvHD [32]. Daclizumab is another humanized mAb against the IL-2 receptor expressed on activated T lymphocytes and has also been evaluated in numerous small studies as well with mixed responses. The majority of these studies demonstrated limited activity and association with an increased incidence of infectious complications [33,34,35,36,37,38].

A smaller study for SR-GvHD using vedolizumab, a T-cell-targeting mAb, noted that patients who progressed early after ruxolitinib failure had significantly worse outcomes, and it was felt this could reflect rapidly progressive GvHD that necessitated early initiation of another line of therapy [39]. Similarly, this pattern of inferior outcomes when ruxolitinib is initiated before CCBT with BAS/INF could suggest that patients already refractory to ruxolitinib would also be less likely to respond to CCBT due to the high severity of GvHD already present. This could arguably also reflect a benefit of starting CCBT early on rather than waiting for ruxolitinib efficacy, particularly in the case of GI GvHD, where parenteral therapy would not face the absorption barriers encountered by oral therapies.

CCBT using BAS/INF is an appealing intervention as it is parenteral and easy to administer compared to oral routes, which is particularly important in GI GvHD. It also does not incur clinically limiting cytopenia like other next-in-line interventions including ruxolitinib, which is substantially important in the first 30–50 days post-transplant when graft dysfunction is common. The utilization of BAS/INF in combination for SR-GvHD has had limited discussion in the literature: only one small 21-patient retrospective analysis using the combination in severe GI GvHD identified 16 patients meeting the criteria for SR-GvHD and showed a high response rate but did not translate to survival benefit [40]. This finding was similar to that seen in other studies of severe GvHD and stresses the importance of prolonged follow-up and the limitations of retrospective and non-randomized design with patients having a mix of pre-transplant diagnoses, conditioning regimens, and degree of HLA matching, all of which are equally pitfalls in our study but perhaps mitigated to some extent by our significantly larger patient population from a single-center cohort.

## 5. Conclusions

CCBT has shown potential efficacy in steroid-refractory aGVHD. The observed ORR of our study is reflective of both advanced and highly refractory aGVHD, with one-third of patients receiving ruxolitinib first and then CCBT as third-line treatment. Given the observed ORR, OS, and NRM differences when ruxolitinib is initiated prior to CBBT, this provides further evidence for CCBT as a suitable second-line alternative or effective salvage therapy for ruxolitinib-refractory GvHD.

## Figures and Tables

**Figure 1 cancers-16-03912-f001:**
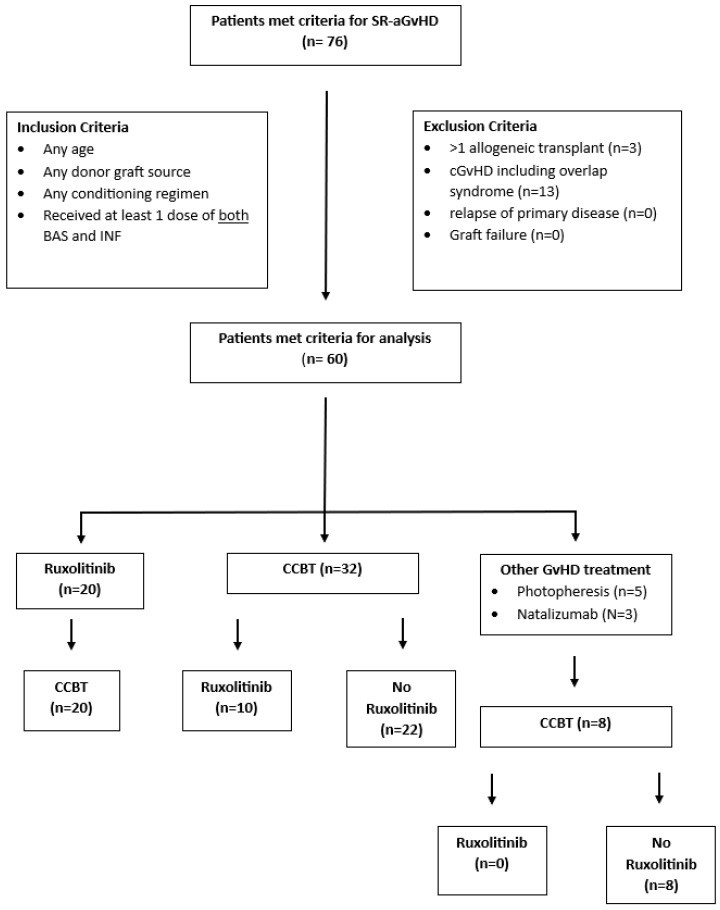
Consort diagram of study patient selection. SR-GvHD: steroid-refractory graft-versus-host disease; BAS: basiliximab; INF: infliximab; CCBT: combined cytokine blockade therapy; GvHD: graft-versus-host disease.

**Figure 2 cancers-16-03912-f002:**
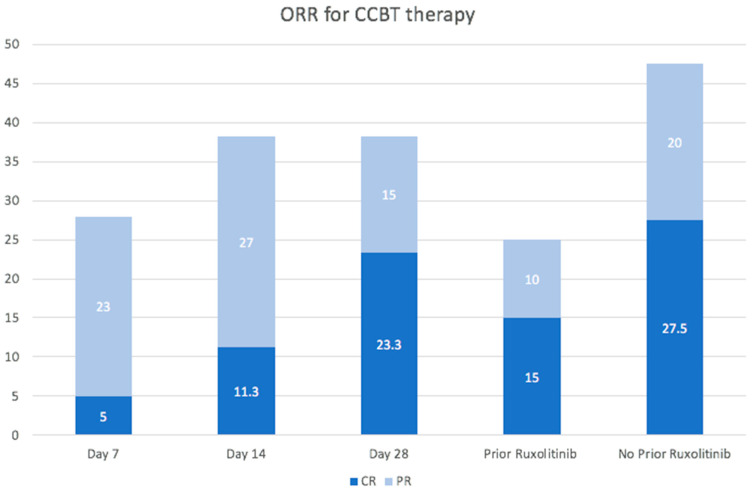
Overall response rate (ORR) for CCBT at days 7, 14, and 28 were 28.3% (17/60; CR 5.0%/PR 23%), 38.3% (23/60; CR 11.3%/PR 27%), and 38.3% (23/60; CR 23.3%/PR 15%), respectively, while patients who received ruxolitinib prior to CCBT had lower ORR at 25% (CR = 15%/PR = 10%) compared to those who did not at 47.5% (CR = 27.5%/PR = 20%).

**Figure 3 cancers-16-03912-f003:**
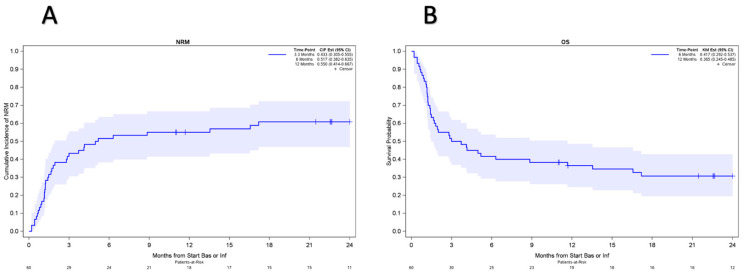
(**A**). Cumulative incidence of NRM was 51.7% (95% confidence interval, 38–64) at 6 months and 55% (95% confidence interval, 41–67) at 12 months. (**B**) OS was 41.7% (95% confidence interval, 29–54) at 6 months and 36.5% at 12 months (95% confidence interval, 26–49).

**Figure 4 cancers-16-03912-f004:**
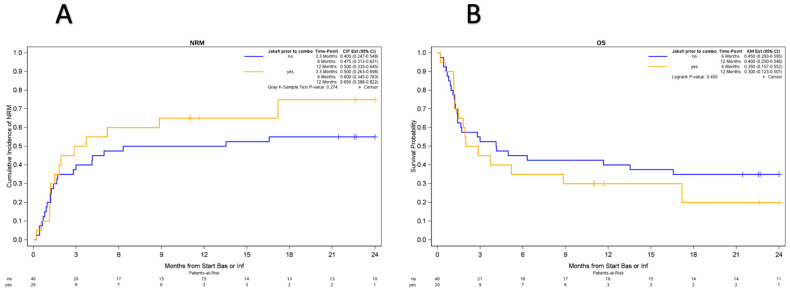
(**A**) NRM was 50% (95% confidence interval, 34–65) at 12 months for those without ruxolitinib prior to CCBT and 65% (95% confidence interval, 39–82) for those with ruxolitinib prior to CCBT. (**B**) OS was 40% (95% confidence interval, 25–55) at 12 months for those without ruxolitinib prior to CCBT and 30% (95% confidence interval, 12–50) for those with ruxolitinib prior to CCBT.

**Table 1 cancers-16-03912-t001:** Baseline characteristics at time of transplant.

	N = 60
Age at HCT, median, range in years	53 (5–78)
Recipient Sex	
Male	39 (65%)
Female	21 (35%)
Race/ethnicity	
White	27 (45%)
Hispanic	20 (33.3%)
Asian	5 (8.3%)
African American	4 (6.7%)
Native American	4 (6.7%)
Diagnosis	
AML	21 (35%)
ALL	9 (15%)
MDS/MPN	14 (23.3%)
Lymphoma	4 (6.7%)
CML/CMML	5 (8.3%)
Non-malignant	5 (8.3%)
Others	2 (3.3%)
Karnofsky performance status	
80–100%	57 (95%)
≤70%	3 (5%)
HCT comorbidity index	
0	11 (18.3%)
1–2	21 (35%)
≥3	28 (46.7%)
Disease risk index	
Non-malignant/other	7 (11.7%)
Low	12 (20%)
Intermediate	23 (38.3%)
High	14 (23.3%)
Very high	4 (6.7%)
Graft source	
Bone marrow	12 (20%)
Cord blood	2 (3.3%)
Peripheral stem cells	46 (76.7%)
Donor source	
HLA-matched related	12 (20%)
HLA- matched unrelated	33 (55%)
HLA-mismatched unrelated	5 (8.3%)
Haploidentical	8 (13.3%)
Cord blood	2 (3.3%)
Gender (donor–recipient)-mismatched HCT	
Female-to-male	11 (18.3%)
Conditioning regimen intensity	
Ablative	26 (43.3%)
Non-myeloablative/reduced intensity	34 (56.7%)
GvHD prophylaxis	
Cyclosporine/MMF x	2 (3.3%)
Cyclosporine /MMF/MTX x	1 (1.7%)
Cyclosporine/MTX x	2 (3.3%)
Tacrolimus/MMF x	1 (1.7%)
Tacrolimus/MMF/cyclophosphamide	11 (18.3%)
Tacrolimus/MTX x	19 (31.7%)
Tacrolimus/MTX/bortezomib	2 (3.3%)
Tacrolimus/sirolimus	21 (35%)
Tacrolimus/sirolimus/ruxolitinib	1 (1.7)

ALL: acute lymphoblastic leukemia; AML: acute myeloid leukemia; CLL, chronic lymphocytic leukemia; CML: chronic myeloid leukemia; CMML: chronic myelomonocytic leukemia; GvHD: graft-versus-host disease; HCT: hematopoietic stem cell transplant; HLA: human leucocyte antigen; MDS, myelodysplastic syndrome; MMF: mycophenolate mofetil; MPN: myeloproliferative neoplasm; MTX: methotrexate.

**Table 2 cancers-16-03912-t002:** GvHD characteristics.

Time to BAS after starting steroid, median days, (range)	7 (−13–63)
Time to INF after starting steroid, median days, (range)	11 (0–150)
Ruxolitinib given pre-CCBT	
No	40 (66.7%)
Yes	20 (33.3%)
Overall aGvHD score at time of CCBT	
Grade II	5 (8.3%)
Grade III	35 (58.3%)
Grade IV	20 (33.3%)
Skin aGvHD stage at time of diagnosis	
Grade 0	32 (53.3%)
Grade 1	6 (10%)
Grade 2	13 (21.7%)
Grade 3	5 (8.3%)
Grade 4	4 (6.7%)
GI aGvHD stage at time of diagnosis	
Grade 0	5 (8.3%)
Grade 1	4 (6.7%)
Grade 2	13 (21.7%)
Grade 3	23 (38.3%)
Grade 4	15 (25%)

aGvHD: acute graft-versus-host disease; BAS: basiliximab; CCBT: combined cytokine blockade therapy; GI: gastrointestinal; INF: infliximab.

## Data Availability

The data that support the findings of this study are available from the corresponding author, [M.A.], upon reasonable request.

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
