# Peer review of "Combined Cytokine Blockade Therapy (CCBT) Using Basiliximab and Infliximab for Treatment of Steroid-Refractory Graft-Versus-Host Disease (SR-GvHD)"

_cancers, 2024, doi:10.3390/cancers16233912_

Round 1

Reviewer 1 Report

Comments and Suggestions for Authors

This is a single-center study on a theme with few previous studies using the same regimen protocol in a critical and individual group of patients, who received bone marrow transplants and are now refractory to steroid treatment to avoid graft versus host disease (GVHD). This study has a limited number of patients recruited compared with the previous more recent ones. The results showed similar data to other cohorts, validating the safety and efficacy of basiliximab and infliximab for in steroid-refractory patients suffering from GVHD.

Reviewer 2 Report

Comments and Suggestions for Authors

The original article deals with second-line anticytokine therapy for steroid-resistant graft-versus-host disease (GVHD) after allogeneic transplantation of hematopoietic stem cells (allo-HCT). Instead of Ruxolitinib, the authors present results of combined pathogenesis-oriented GVHD therapy with Basiliximab and Infliximab, well-known anti-inflammatory monoclonal antibodies that showed clinical effects in previous studies. The authors conclude that such combined approach can provide successful salvage therapy in approximately 1/3 of patients with steroid- and ruxolitinib-refractory GVHD. This original study seems to be of sufficient clinical significance. 

However, some issues could be specified.

Remarks

Abstract and Introduction: The aim and endpoints of the study should be clearly highlighted.

Line 88: …any age… - numerical data should be provided here from Table 1, e.g., '53 (5-78) y.o.'

Line 89: the GVHD cases were collected over 10 years. In results or Discussion, one should mention the prolonged collection time, since primary GVHD prophylaxis was changed over this period (e.g., post-transplant cyclophosphamide).

Line 137 and Results: of sufficient interest would be additional data about incidence of serious infections complications (local or septicemia) in these patients, compared with glucocorticoids or Ruxolitinib treatment.

Line 167 (Section 3.1., CCBT Response). The authors should clearly define the control (or comparison) group observed during the 10-year period, in order to show significant benefit of the CCBT approach in treatment of refractory GVHD.

Line 200: Fig 4. What is the potential cause of higher NRM/ lower OS in Ruxo-pre-treated patients: either general GVHD severity, or some additive effects of Jakafi?  It may be discussed in more details. 

Discussion

Line 235-238: one should mention which Mab (Basiliximab and Infliximab) seems to be the principal therapeutic agent in this dual treatment scheme.

Some limitations of the study (e.g., long-time recruiting of the patients) should be indicated. 

Few stylish issues and misprints are found in the text:

Line 142: Meir – correct to Meier

Line 235: … in isolation…   better: …as monotherapy…

Line 271: …ruxolitinib… refractory  please make  …ruxolitinib-refractory…

Minor copy-editing is required

Comments on the Quality of English Language

MInor copy-editing is recommended

Reviewer 3 Report

Comments and Suggestions for Authors

The authors in this study have evaluated overall response rate, non-relapse mortality an overall survival associated with CCBT. The study shows CCBT has potential efficacy in steroid refractory GI aGvHD and given observed ORR when used as second line.

However, the authors need to work on the mentioned comments to make the manuscript comprehensive and publishable in the Cancers, MDPI.

1. The percent match (plagiarism) should be less than 20%.

2. The introduction needs to be improved and add latest references.

3. The figures, charts and tables needs better presentations and resolution( higher pixels).

4. The survival curves needs better presentation and resolutions. Add p-values and HR ratios and better legend presentations.

5. The English and scientific language needs to be improved and text editing is needed to improve presentation.

Comments on the Quality of English Language

The English and scientific language needs to be improved and text editing is needed to improve presentation.
